# Purification of recombinant bacterial collagens containing structural perturbations

Sonal Gahlawat [1], Vikas Nanda [2,3], David I. Shreiber [1] *

1 Department of Biomedical Engineering, Rutgers, The State University of New Jersey, Piscataway, NJ, United States of America, 2 Department of Biochemistry and Molecular Biology, Robert Wood Johnson Medical School, Rutgers, The State University of New Jersey, Piscataway, NJ, United States of America, 3 Center for Advanced Biotechnology and Medicine, Rutgers, The State University of New Jersey, Piscataway, NJ, United States of America

* shreiber@soe.rutgers.edu

## Abstract

*Streptococcus pyogenes*-derived recombinant bacterial collagen-like proteins (CLPs) are emerging as a potential biomaterial for biomedical research and applications. Bacterial CLPs form stable triple helices and lack specific interactions with human cell surface receptors, thus enabling the design of novel biomaterials with specific functional attributes. Bacterial collagens have been instrumental in understanding collagen structure and function in normal and pathological conditions. These proteins can be readily produced in *E. coli*, purified using affinity chromatography, and subsequently isolated after cleavage of the affinity tag. Trypsin is a widely used protease during this purification step since the triple helix structure is resistant to trypsin digestion. However, the introduction of Gly→X mutations or natural interruptions within CLPs can perturb the triple helix structure, making them susceptible to trypsin digestion. Consequently, removing the affinity tag and isolating collagen-like (CL) domains containing mutations is impossible without degradation of the product. We present an alternative method to isolate CL domains containing Gly→X mutations utilizing a TEV protease cleavage site. Protein expression and purification conditions were optimized for designed protein constructs to achieve high yield and purity. Enzymatic digestion assays demonstrated that CL domains from wild-type CLPs could be isolated by digestion with either trypsin or TEV protease. In contrast, CLPs containing Gly→Arg mutations are readily digested by trypsin while digestion with TEV protease cleaved the His6-tag, enabling the isolation of mutant CL domains. The developed method can be adapted to CLPs containing various new biological sequences to develop multifunctional biomaterials for tissue engineering applications.

## Introduction

Over the past two decades, recombinant bacterial collagen from *Streptococcus pyogenes* has evolved into an indispensable biomaterial to gain insight into the sequence-structure-function relationship of normal and pathological collagens. *S. pyogenes* produces collagen-like

---

**Data Availability Statement:** All relevant data are within the manuscript and its Supporting Information files.

**Funding:** This research was supported by the New Jersey Health Foundation (Grant/Award Number:

PC101-20, https://www.njhealthfoundation.org) and The Marfan Foundation (Grant/Award Number: 127611, https://marfan.org) awarded to D.I.S. The funders had no role in study design, data collection and analysis, decision to publish, or preparation of the manuscript.

**Competing interests:** The authors have declared that no competing interests exist.

proteins (also known as Scl2 or CLPs) that contain an N-terminal signal sequence, a variable globular domain (V-domain), a collagen-like (CL) triple helix domain consisting of repeating units of $(Gly-X-Y)_n$, and a C-terminal cell wall attachment domain [1]. The V-domain contains a coiled-coil motif and is essential for the trimerization and proper folding of the triple helix [2]. The recombinant CLPs can self-assemble into a stable triple helix structure even in the absence of hydroxyproline residues. The high content of proline residues along with charged and polar residues contribute to the stability of the triple helix via electrostatic interactions [3, 4]. As such, recombinant CLPs exhibit a high thermal stability of 35–39˚C, comparable to animal collagen [5, 6]. Unlike animal collagens, CLPs do not inherently bind to human cell surface receptors and other components of the extracellular matrix (ECM) to elicit specific responses [7, 8]. Consequently, they can provide an effective biological control, allowing selective integration of specific human collagen ligand-binding sites to study specific protein-collagen interactions via recombinant DNA technology [9–13]. Hence, engineered CLPs can be used to probe the structure and function of animal collagen, including single amino acid substitutions and natural interruptions to mimic disease-causing mutations in collagen disorders.

Recombinant CLPs can be easily expressed in *E. coli* to achieve high protein expression using a cold-shock system that selectively induces expression of the target protein at low temperatures [14]. One of the most common methods to purify these bacterial CLPs is to include a polyhistidine (His6) tag at the N-terminal of the protein, followed by protein purification using immobilized metal affinity chromatography (IMAC) that utilizes the interaction between His6-tagged CLP and immobilized $Ni^{2+}$ ions supplied by the chromatography matrix. Previous applications of CLPs to study triple helix structure and function all included a trypsin protease site to remove the His6-tag to obtain CL domains from the full-length designed construct [3, 10, 11, 13, 15–18]. This was motivated by the established proteolytic resistance of a native collagen triple helix to digestion by trypsin, pepsin, and chymotrypsin (**Fig 1**) [3, 5, 8, 16, 19]. However, a triple helix with disruptions and a partially folded structure is sensitive to digestion [20, 21]. Therefore, recombinant CLPs containing natural interruptions and mutations that could potentially disrupt the triple helical structure become susceptible to trypsin digestion (**Fig 1**) [22–27]. As a result, it is very challenging to remove the His6-tag or any other affinity tag without disrupting the triple helix structure of the proteins, and researchers who have used CLPs to study these kinds of interruptions forgo the cleavage step and leave the His6-tag intact.

We developed an efficient and reproducible purification protocol to isolate recombinant CLPs with structural perturbations by incorporating a Tobacco Etch Virus (TEV) protease cleavage site. Structurally distorted triple helices were generated through the introduction of Gly→Arg mutations within the integrin-binding site of CL domains. The Gly→Arg mutation would provide differential folding effects and has been associated with collagen disorders, including Osteogenesis Imperfecta and Ehlers-Danlos Syndrome and its subtypes (https://eds.gene.le.ac.uk/). The TEV protease is responsible for cleaving the His-tag from the designed full-length construct, while trypsin can be used to investigate the structural integrity of the triple helix (**Fig 1**). Optimization of protein expression and purification was performed to maximize bacterial CLPs production. Several parameters were examined that could affect protein expression, including bacterial host strain, culture media, and inducer concentration. The results indicate that this method can be applied to express and purify recombinant CLPs with sequences of varied molecular flexibility to obtain CL domains for downstream applications, including biophysical characterization of mutant proteins and study of cell function and behavior due to the included mutation(s).

A. Intact triple helix

B. Disrupted triple helix

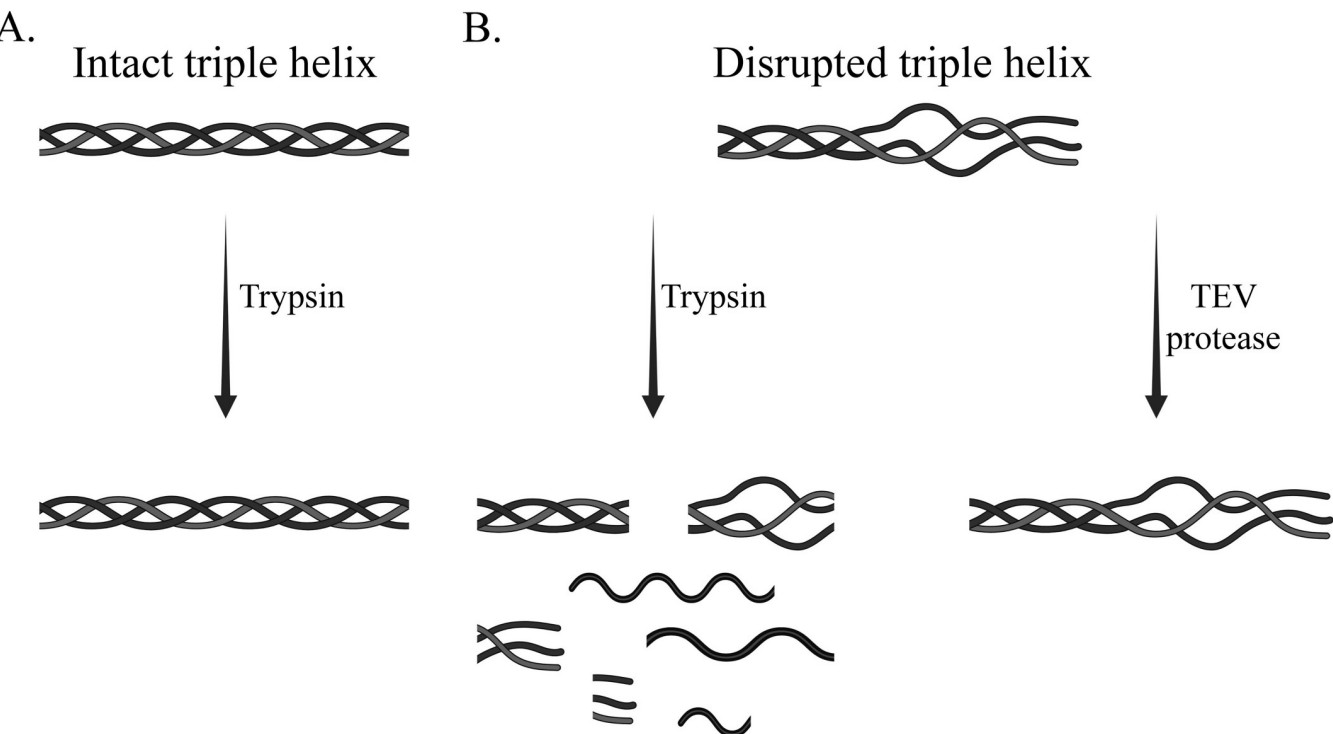

**Fig 1. Schematic illustration of disrupted triple helix digestion by trypsin and TEV protease.** (A) An intact triple helix is resistant to digestion by trypsin. (B) Due to mutations and natural interruptions, the triple helix become susceptible to trypsin digestion, which might result in protein degradation. In contrast, TEV protease does not degrade disrupted triple helix, enabling the isolation of mutant triple helices for more precise characterization and functional studies and for use as a potential biomaterial.

## Materials and methods

### Chemicals and reagents

Sodium phosphate monobasic (AC389870025), sodium phosphate dibasic (S374-500), imidazole (AC122025000), carbenicillin (BP26485), tryptone (BP1421-2), yeast extract (BP9727-2), and Luria Broth (LB) Agar (BP9724-500) were purchased from Fisher Scientific. Dialysis tubing cellulose membrane (D9527-100FT), acetic acid (695092–2.5L), trypsin (T1426-50MG), and Amicon® Ultra-15 Centrifugal Filter Unit (UFC901024) were purchased from Millipore Sigma. Sodium Chloride (0241–2.5KG) was purchased from VWR. Ni -NTA Agarose (30210) was purchased from Qiagen. 4X Laemmli sample buffer (1610737), Precision Plus protein dual color standard (1610374), 4–15% Mini-PROTEAN® TGX™ Precast Protein Gels (4561086), and 2-mercaptoethanol (1610710) were purchased from Bio-Rad Laboratories. BL21-DE3 competent cells (C2527I) were purchased from New England BioLabs. Isopropylthio-β-galactoside, IPTG, (I2481C50) was purchased from Gold Biotechnology. Collagen-I (C857) was purchased from Elastin Products Company, Inc. pCold-I expression vector was purchased Takara Bio Inc.

### Design of recombinant bacterial CLPs

The gene constructs used for bacterial CLPs are based on the DNA sequence for the fragment of the Scl2.28 allele of *S. pyogenes* encoding for the combined globular and collagen-like proteins but lacking the C-terminal domain, as previously described [1, 3, 5, 8, 24]. In brief, constructs included an N-terminal globular domain (V) followed by two CL domains. An enzymatic cleavage site was introduced between the V-domain and the two following CL

domains to facilitate isolation of the CL domains. Constructs contained either the trypsin cleavage sequence (LVPR↓GSP) or the TEV protease sequence (ENLYFQ↓G). This DNA sequence, termed $VCL_2$, was synthesized commercially with codon optimization for expression in *E. coli* (GenScript, Piscataway, New Jersey), and served as the template for creating the mutant construct with a Gly-to-Arg mutation at residue #201 and #444 within the IBS of each CL domain, which was denoted $VCL_2(G \rightarrow R)$. The sequences of the final constructs were confirmed by sequencing prior to transformation and protein expression. The final DNA sequences of all recombinant bacterial collagens were cloned into the pCold-I vector systems (GenScript, Piscataway, New Jersey) for expression in *E. coli*. The constructs also included an N-terminal His6-tag that was provided by the pCold-I vector for protein purification.

## Expression and purification of the proteins

All constructs were transformed into *E. coli* BL21-DE3 strain. For protein expression, a positive clone was selected for the primary culture and was grown in LB media with 100 μg/mL ampicillin at 37˚C with shaking at 250 RPM for overnight. The following day, the primary culture was used to inoculate 1L of 2X YT medium (10 gm/L yeast extract, 16 gm/L tryptone, and 5 gm/L NaCl, pH 7.4) with ampicillin (100 μg/mL). The cells were grown at 37˚C with shaking until the A600 absorbance reading reached an optical density in the range of 0.4 to 0.6 A.U. Subsequently, the cells were cooled down to 25˚C for 1 hour, and 0.1 mM isopropyl-β-D-thio-galactopyranoside (IPTG) was added for protein induction. After 10 hours of protein induction, the cells were further cooled to 15˚C for 14 hours, after which the cells were harvested by centrifugation (4,000 RPM, 20 mins, 4˚C). The cell pellet was stored at -80˚C until further use.

For protein extraction, the cell pellet was thawed and resuspended in lysis buffer, and the cells were ruptured by sonication in ice water for 10 mins (30 sec ON and 30 sec OFF cycle at 40% amplitude). Clarified lysate was obtained by centrifugation at 18,000 RPM for 30 mins at 4˚C. The His-tagged recombinant proteins were purified by immobilized metal affinity chromatography (IMAC) using $Ni^{2+}$ charged affinity resin (Ni-NTA, Qiagen, Germantown, Maryland). To aid in the binding of $Ni^{2+}$ with His-tagged protein, the clarified lysate was incubated with Ni-NTA, pre-equilibrated with binding buffer with gentle shaking at 4˚C. (NOTE: for a cell pellet obtained from 500 mL bacterial culture, 2 gravity columns were used for the purification with each column containing approximately 5 mL of Ni-NTA resin.) After an hour of incubation, the column was washed with (i) 20 column volumes (CV) of binding buffer (50 mM sodium phosphate buffer, 500 mM of sodium chloride, and 10 mM of imidazole, pH 8), (ii) 20 CV of high-salt buffer (50 mM sodium phosphate buffer, 1 M of sodium chloride, and 10 mM of imidazole, pH 8), and (iii) 15–20 CV of low imidazole buffer (50 mM sodium phosphate buffer, 500 mM of sodium chloride, and 50 mM of imidazole, pH 8) to remove the non-specific proteins bound to the resin. Finally, the protein was eluted in a step gradient of imidazole concentration (100 mM, 250 mM, and 500 mM and pH 8) with three fractions (10 mL each) collected for each concentration. The purification buffers contained β-mercaptoethanol (BME) as the reducing agent to prevent any disulfide bonds formation. All fractions containing the protein were collected and verified for purity level with SDS-PAGE. Eluted fractions containing the protein were pooled and dialyzed against 5 mM sodium phosphate buffer, pH 7.4 for 48 hours with three buffer changes. The dialyzed protein was lyophilized and stored at -80˚C.

## Enzymatic digestion assay

Enzymatic digestion assays were performed to examine the structural stability of triple helix. All recombinant proteins were digested with either trypsin or TEV protease at a ratio of 1:25

and 1:10, respectively, at 20˚C for 2 mins, 5 mins, 15 mins, or overnight (16 hours). After digestion, the reaction was stopped by adding 4X SDS sample buffer (3:1 ratio) and heating the samples to 95˚C for 5 mins. The digestion profile of the constructs was analyzed by SDS-PAGE.

## Results

### Design and expression of recombinant constructs with trypsin and TEV protease sites

The control protein, referred to as $VCL_2$, includes an N-terminal V-domain and two tandem CL domains ($CL_2$), which are separated by the linker sequence GAAGVM (**Fig 2A**). Each of the two CL domains included an integrin binding site (IBS), GFPGER, integrated within the triple helix structure for cell binding. GGPCPPC sequences were inserted at the N- and C-terminal of the CL domains to aid in stability and allow subsequent functionalization for downstream applications. The $VCL_2(G{\rightarrow}R)$ mutant protein includes a single Gly→Arg mutation (GFP**R**ER) near the middle of the CL domain within the IBS sequence to have minimum impact on the folding and assembly of the triple helix (**Fig 2B**).

Lastly, the introduction of two different enzymatic cleavage sites between the V-domain and $CL_2$ domains allowed us to remove His6-tag and isolate CL domains from the full-length

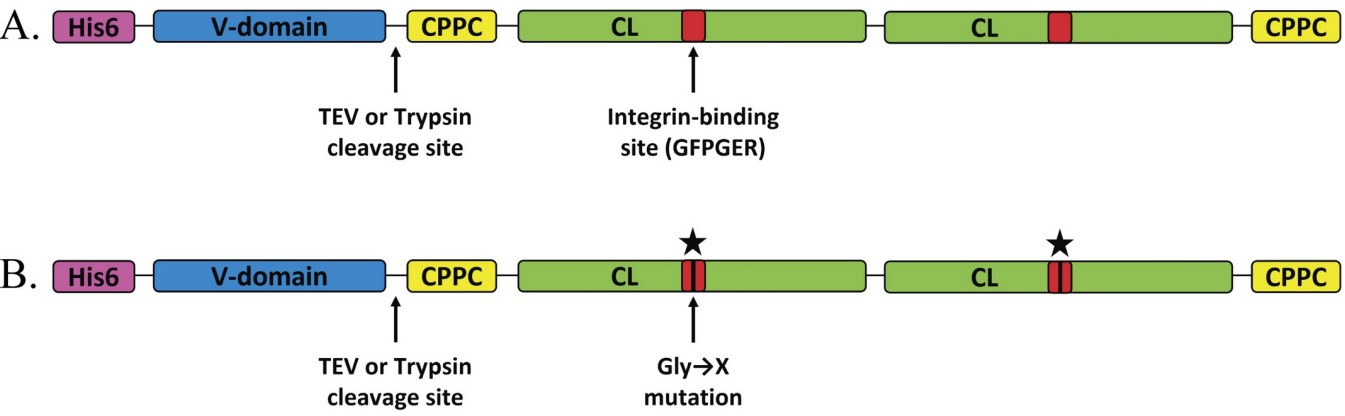

| Construct Name | Description |
|---|---|
| $VCL_2$-Trypsin | Wild-type protein containing a trypsin protease cleavage sequence |
| $VCL_2$-TEV | Wild-type protein containing a TEV protease cleavage sequence |
| $VCL_2(G{\rightarrow}R)$-Trypsin | Mutant protein containing a Gly→Arg mutation within the IBS and a trypsin protease cleavage sequence |
| $VCL_2(G{\rightarrow}R)$-TEV | Mutant protein containing a Gly→Arg mutation within the IBS and a TEV protease cleavage sequence |

**Fig 2. Schematic illustration of the design of recombinant bacterial collagens.** (A) $VCL_2$ with a protease cleavage site (highlighted by "↑") and an IBS within each CL domain. (B) Version of the $VCL_2$ construct with Gly→Arg within the IBS, where the second Gly residue of GFPGER was mutated to Arg (highlighted by "★"). The 'CPPC' domains represent the amino acid sequences inserted at the N- and C- terminal of the CL domains to aid in stability. (C) Description of the designed protein constructs.

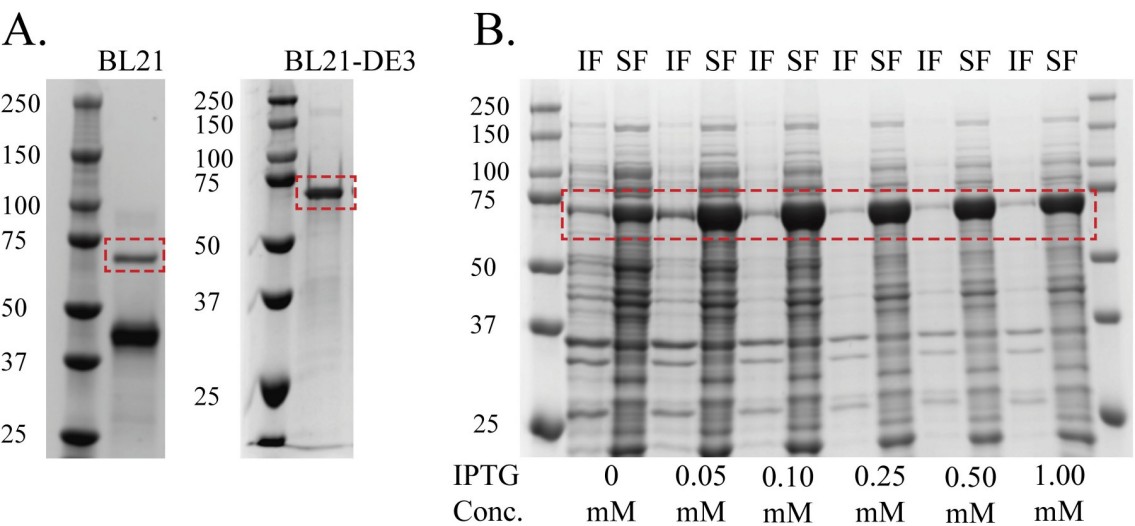

**Fig 3. Protein expression of VCL$_2$ was optimized to achieve high expression and protein solubility.** (A) SDS-PAGE of the VCL$_2$ purified proteins expressed in *E. coli* BL21 and BL21-DE3. Protein expression in BL21-DE3 cells resulted in purification of VCL$_2$ with high purity, without any degradation products. (B) SDS-PAGE of insoluble fractions (IF) and soluble fractions (SF) of VCL$_2$ after induction with varying IPTG concentrations, where lower IPTG concentrations induced high protein expression. Molecular Weight standards are in kDa. Conc. = Concentration.

constructs. Four different types of constructs were designed for protein expression and purification: VCL$_2$-Trypsin, VCL$_2$-TEV, VCL$_2$(G→R)-Trypsin, and VCL$_2$(G→R)-TEV (**Fig 2C**).

We used the cold-shock pCold-I vector system to express and purify recombinant CLPs with and without specific Gly→X mutations. Recombinant protein overexpression at low temperatures often improves protein solubility and stability [14]. We optimized the protein expression by comparing different *E. coli* strains (BL21 vs. BL21-DE3), culture media (TB vs. 2X YT), and inducer concentration (0–1 mM). We initially utilized the BL21 bacterial strain for the expression of the VCL$_2$ protein, since protein expression was driven by the cold-shock promoter, unlike a traditional T7 RNA polymerase-based vector, where protein expression is strictly controlled by the inducer. Purification of BL21-expressed protein resulted in target protein and a dominant lower molecular weight band that is likely a CL$_2$ degradation product (**Fig 3A**). By switching to BL21-DE3 bacterial strain, we nearly eliminated the second band (**Fig 3A**).

To improve expression and solubility, we tested different IPTG concentrations. High protein expression was obtained using lower IPTG concentrations (0.05 and 0.01 mM, **Fig 3B**) and inducing protein expression at the mid-exponential growth phase using 2X YT media vs. TB media. Protein expression using TB media led to high protein loss in inclusion bodies, which could not be recovered using detergents. To identify the optimal conditions for protein expression, we selected conditions that led to high protein expression in the soluble fraction and a low expression in the insoluble fraction. Testing various IPTG concentrations allowed us to identify an optimal level of inducer for protein expression with minimal protein loss in inclusion bodies. The combination of low temperatures and lower IPTG concentrations enhanced both protein expression and solubility.

A bench-scale purification of all constructs was carried out at 4°C. The His6-tag at the N-terminal allowed protein purification by IMAC using a Ni-NTA resin, resulting in pure samples for each protein construct as verified by SDS-PAGE (**Fig 4**). We also utilized an imidazole gradient to obtain a high yield with maximum purity. We used three concentrations of imidazole (100 mM, 250 mM, and 500 mM) to elute the protein and determine the optimal

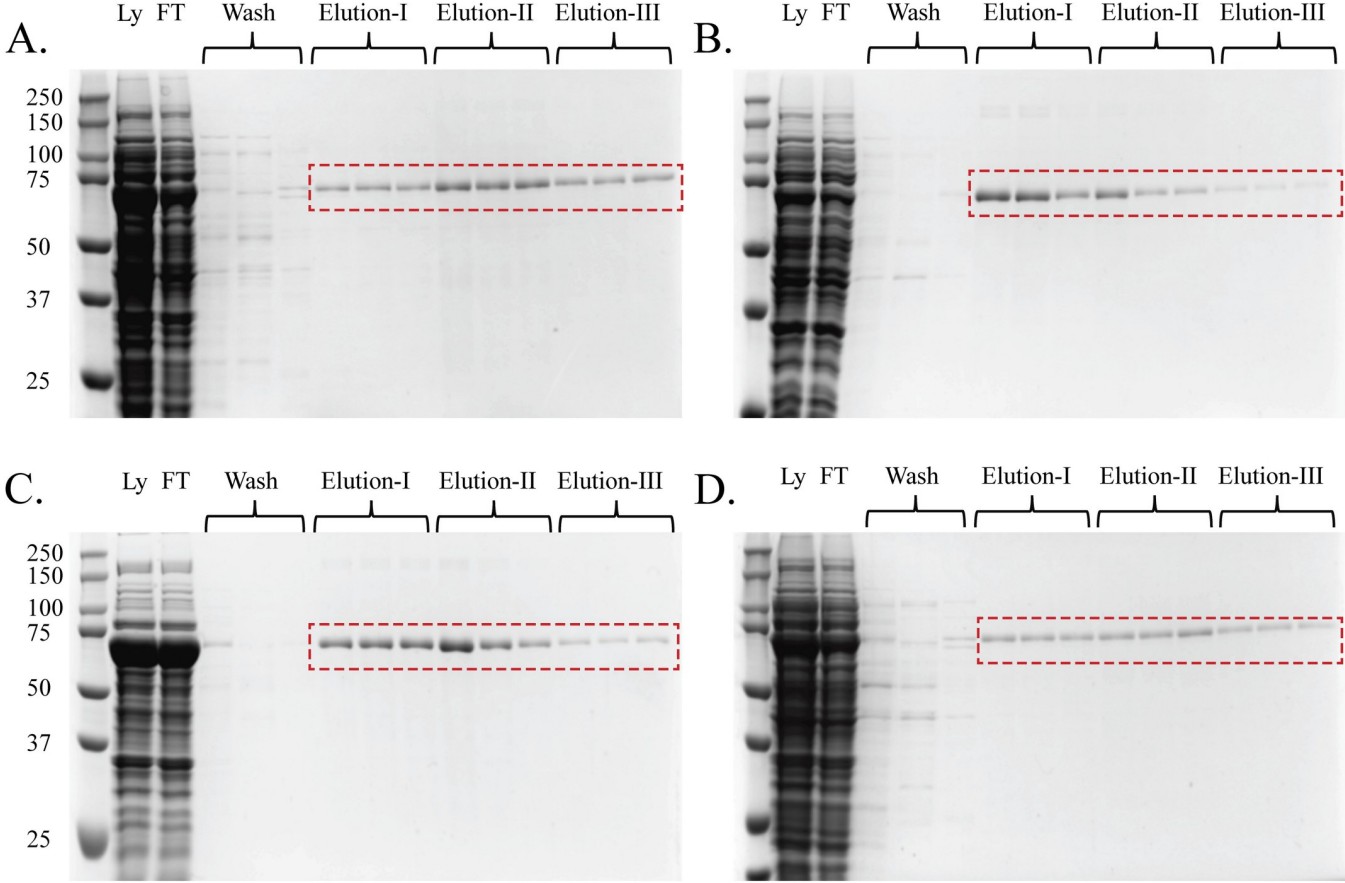

**Fig 4. Ni-NTA purification of expressed proteins lead to high purity.** SDS-PAGE of the purified proteins, (A) VCL$_2$-Trypsin, (B) VCL$_2$-TEV, (C) VCL$_2$(G→R)-Trypsin, and (D) VCL$_2$(G→R)-TEV. Cell lysate was incubated with Ni-NTA resin and FT was collected. The column was washed with binding buffer, high salt buffer, and low imidazole buffer to remove non-specific bound proteins. His-tagged protein was eluted with buffer containing 100 mM (Elution-I), 250 mM (Elution-II), and 500 mM (Elution-III) imidazole with three fractions collected per concentration. Molecular weight standards in kDa, Ly = lysate, and FT = Flowthrough (containing the unbound protein).

concentration for elution of the protein. Three different fractions were collected for each imidazole concentration to separate contaminants not removed during washing and to achieve full recovery of the target protein.

The monomeric molecular weight of VCL$_2$-Trypsin, VCL$_2$-TEV, VCL$_2$(G→R)-Trypsin, and VCL$_2$(G→R)-TEV are 57.24 kDa, 57.41 kDa, 57.55 kDa, and 57.61 kDa, respectively. On SDS-PAGE, the protein bands for monomeric chains were higher than expected, which has previously been shown for proteins with rod-like structures [3]. The protein purity was sufficient to examine the effectiveness of each protease to isolate CL domains with and without Gly→X mutations.

High protein expression in all constructs was seen, as demonstrated by the high-intensity band in loaded lysates on the SDS-PAGE. Such high protein expression led to inefficient binding of the protein with Ni-NTA resin, resulting in protein elution during sample loading, despite a variety of attempts to improve the binding efficiency, including optimization of the binding buffer with varying concentrations of imidazole and detergents, purification under denaturing conditions, changing the incubation time between the protein and resin, and increasing the volume of the resin. Recovery of the protein lost in the flow-through was easily achieved by performing multiple rounds of purification using the collected flow-through.

Alternatively, larger-sized gravity columns or high-performance liquid chromatography (HPLC) instruments can be utilized to attain high yields in one-step purification.

## Structural perturbation in triple helix increases susceptibility to trypsin digestion

Trypsin digestion assays are one of the most common techniques employed to examine the structural stability of triple helices. Mutations that disrupt the triple helix conformation can increase proteolytic susceptibility. As expected, trypsin digestion of $VCL_2$-Trypsin cleaves the His6-tag and V-domain, allowing the isolation of $CL_2$ protein (**Fig 5A**). The presence of an extra band is observed, which suggests digestion of a small fraction of unfolded $CL_2$. However, the introduction of Gly→Arg within the IBS of bacterial collagen, $VCL_2(G→R)$-Trypsin, increased sensitivity to trypsin digestion with detectable $CL_2$ cleavage products after only 2 minutes (**Fig 5B**). This is consistent with the highly disruptive nature of Gly→Arg substitutions within the triple helix [22, 28–30]. After overnight digestion with trypsin, the $VCL_2(G→R)$-Trypsin protein is entirely degraded, hampering purification and significantly reducing the final yield of $CL_2(G→R)$.

## TEV protease sites facilitate purification of CL domains containing structural perturbations

To minimize proteolysis of $CL_2(G→R)$ during purification, we synthesized $VCL_2$ and $VCL_2(G→R)$ constructs containing the TEV protease cleavage site. The digestion profile of $VCL_2$-TEV with TEV protease was similar to that of $VCL_2$-Trypsin digested with trypsin (**Fig 5C**). While trypsin digestion of $VCL_2(G→R)$-Trypsin produced multiple degradation products, TEV protease digestion of $VCL_2(G→R)$-TEV only produced the His6-tag + V-domain and the target $CL_2(G→R)$ (**Fig 5D**). As expected, trypsin digestion of $VCL_2(G→R)$-TEV produced multiple degradation products (**Fig 5E**). These results confirm that the purification of intact CL domains with G→X mutations can be achieved by replacing trypsin with TEV protease. Replacing the trypsin cleavage site with TEV offers a reliable method to isolate CL domains containing structural perturbations irrespective of the type of substituted amino acid, the location of the mutation, and the surrounding local sequence, all of which play a decisive role in determining the conformation and stability of the triple helix.

## Discussion

Given that CLPs are increasingly used in collagen-based biomaterials and as tools for understanding collagen structure and function [31–36], it is critical to develop effective methods for purifying and studying them. The inclusion of the His6-tag at the N-terminal sequence allowed the purification of expressed CLPs using IMAC, which can be easily scaled up. However, isolating CL domains and removing the His6-tag (or any other affinity tag) using current protocols depends on the formation of a fully-folded, triple helix structure.

We have developed a robust protocol for isolating full-length recombinant CLPs containing structural perturbations due to Gly→X mutations. Bacterial CLPs are often engineered to incorporate specific sequences of interest, such as Gly→X mutations and other natural interruptions, to study collagen's structure and function, especially in inherited collagen disorders. Such modifications could also be used to alter biomaterial mechanics or tailor cell-biomaterial interactions. While existing protocols based on trypsin digestion provide a reliable tool to investigate the impact of Gly substitutions on triple helix structure, the use of the same enzyme for purification restricts our ability to obtain full-length recombinant collagens containing Gly substitutions.

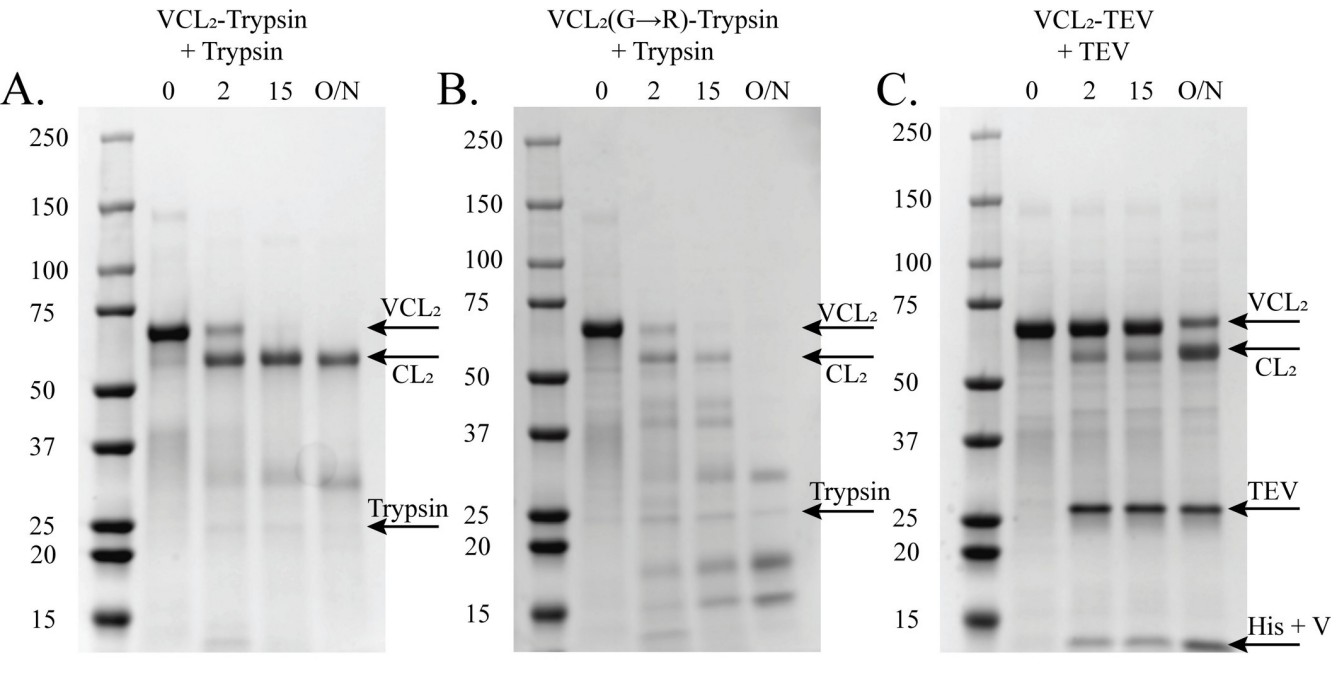

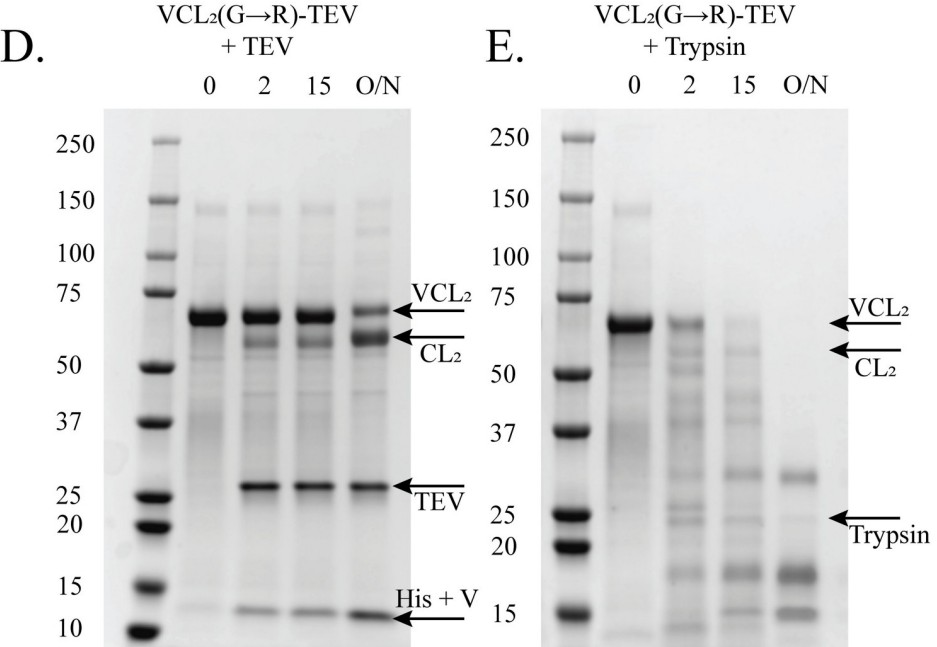

**Fig 5. Incubation of distorted triple helices with trypsin demonstrated protein digestion while incubation with TEV protease allowed isolation of CL domains containing Gly→X mutations.** SDS-PAGE of (A) VCL$_2$-Trypsin, (B) VCL$_2$(G→R)-Trypsin, (C)VCL$_2$-TEV, (D) VCL$_2$(G→R)-TEV, and (E) VCL$_2$(G→R)-TEV purified proteins digested with either trypsin or TEV protease for different time points (0 mins, 2 mins, 15 mins, and O/N). Digestion of VCL$_2$ with trypsin and TEV protease resulted in removal of His6-tag and V-domain and isolation of CL$_2$. However, digestion of VCL$_2$(G→R) with trypsin led to protein digestion starting at 2 mins only. Digestion of VCL$_2$(G→R)-TEV with TEV protease resulted in the isolation of CL$_2$(G→R) protein without degrading the protein. Molecular weight standards in kDa, O/N = Overnight digestion with the protease.

In this study, we identified shake flask culture conditions that maximize bacterial CLP expression. Selecting a suitable *E. coli* strain proved an essential step since the type of strain had a major impact on protein expression, solubility, and yield. Moreover, culture media

composition and conditions affect bacterial growth, which will ultimately impact protein expression and solubility. By sampling culture media and inducer concentration, we were able to obtain high protein yields. Furthermore, to prevent unwanted degradation products, we incorporated a TEV protease cleavage site instead of the widely-used trypsin site to remove the His6-tag and V-domain. This enabled the isolation of the CL domains containing Gly→Arg mutation within the IBS of CL domain. This method can be further adapted to engineering CLPs containing multiple G→X mutations, ligand-binding sites, natural interruptions, or other sequences of interest that are found in animal collagens that are anticipated to compromise triple helix integrity.

## Conclusions

While trypsin is a widely used protease to extract and purify animal collagens for commercial applications, low-substrate specificity and the promiscuous nature of the protease suggest a high potential for mis-cleavages at undesired locations, resulting in irreproducible and unpredictable results. This is particularly important when designing a tailored biomaterial using recombinant bacterial collagens for biomedical applications. Human collagen consists of regions with variable flexibility, which play a crucial role in imparting structural and functional properties in their native environment. Engineering CLPs with specific biological functionality, including cell-cell and cell-ECM interactions or cell-guided degradation, requires incorporating human collagen sequences that can alter the triple helix structure of these fabricated biomaterials. Thus, our method offers one approach to introduce a wide range of sequences to develop advanced biomaterials for tissue engineering applications. This technology provides a versatile platform to design novel biomaterials, where the introduction of multiple biological motifs could potentially impact the triple helix structure.

## Supporting information

**S1 File.**
(PDF)

## Acknowledgments

The authors acknowledge Natalie Losada and Dr. Eddy Arnold (Center for Advanced Biotechnology and Medicine, Rutgers University) for providing access to protein purification resources and Jennifer Timm (Department of Marine and Coastal Sciences and Center for Advanced Biotechnology and Medicine, Rutgers University) for providing valuable expertise in protein purification methodology and sharing the TEV protease.

## Author Contributions

**Conceptualization:** Sonal Gahlawat, Vikas Nanda, David I. Shreiber.

**Data curation:** Sonal Gahlawat.

**Formal analysis:** Sonal Gahlawat.

**Funding acquisition:** David I. Shreiber.

**Investigation:** Sonal Gahlawat.

**Methodology:** Sonal Gahlawat, Vikas Nanda, David I. Shreiber.

**Resources:** Vikas Nanda, David I. Shreiber.

**Supervision:** Vikas Nanda, David I. Shreiber.

**Writing – original draft:** Sonal Gahlawat, David I. Shreiber.

**Writing – review & editing:** Sonal Gahlawat, Vikas Nanda, David I. Shreiber.

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
