## [Decision Letter · Decision Letter 0]

23 Mar 2023

PONE-D-23-06762Purification of recombinant bacterial collagens containing structural perturbationsPLOS ONE

Dear Dr. Shreiber,

Thank you for submitting your manuscript to PLOS ONE. After careful consideration, we feel that it has merit but does not fully meet PLOS ONE’s publication criteria as it currently stands. Therefore, we invite you to submit a revised version of the manuscript that addresses the points raised during the review process.

We look forward to receiving your revised manuscript.

Kind regards,

Yong Wang

Academic Editor

PLOS ONE

Journal Requirements:

Additional Editor Comments (if provided):

Comments from Reviewer#1:

This manuscript by Gahlawat et al described a simple laboratory effort to purify a collagen like protein (CLP) having Gly-to-X mutations that was first expressed as a fusion protein. They provided evidence showing protease TEV is better suited for the removal of the His-tagged V-domain than trypsin because of the folding problem related to the mutation. Yet, there is no data and no mention on the folding of the peptides. Did the digestion act on monomer during purification or on folded trimer? Can the triple helix form without the removal of V-domain? If not, how can they claim the varied sensitivity of the fusion protein to proteases is related to the ‘partially folded structure’?

Other major problems:

Figure 1 is missing.

The optimization experiments are notoriously difficult to reproduce. The authors did not mention the reproducibility of the results.

Overall, the work felt like an incomplete effort and did not have enough reproducible results to meet the requirement of a research article.

Comments from Reviewer#2:

The authors present an alternative method for purifying recombinant bacterial collagens (Collagen-like proteins, CLP) containing structural perturbations or more specifically the mutation Gly to Arg. The motivation for developing the purification method is that the widely used purification method, which uses trypsin digestion to remove the affinity tag, also degrade the structural altered CLP. Instead of trypsin, TEV protease and TEV protease cleavage is used for removal of the affinity tag.

In total the authors make four constructs with TEV or Trypsin cleavage sites combined with native CLP or CLP containing the Gly-Arg mutations. By blabla and enzymatic digestion assay, they demonstrate that CLPs containing Gly→Arg mutations are readily digested by trypsin while digestion with TEV protease only cleaved the affinity tag.

The manuscript would benefit from:

• Specify the rationale for choosing to substitute Glycine with Arginine

• Indicate the number of the amino acid substitution or specify which glycine was substituted in the integrin binding site.

• Reference for the contribution of the mutation Gly to Arg change the structure of CLP

• Reference or argument that Gly to X would have same effect as Gly to Arg

• Figure 1:

o Panel C: Missing figure text.

o Panel A:

Exchange “Protease cleavage sites” with “TEV or Trypsin cleavage site”.

Specify which glycine was mutated in the integrin binding site

• Line 79: forgo  forgot

Reviewers' comments:

Reviewer's Responses to Questions

**Comments to the Author**

1. Is the manuscript technically sound, and do the data support the conclusions?

Reviewer #1: Yes

Reviewer #2: No

2. Has the statistical analysis been performed appropriately and rigorously? 

Reviewer #1: Yes

Reviewer #2: No

3. Have the authors made all data underlying the findings in their manuscript fully available?

Reviewer #1: Yes

Reviewer #2: Yes

4. Is the manuscript presented in an intelligible fashion and written in standard English?

Reviewer #1: Yes

Reviewer #2: Yes

5. Review Comments to the Author

Reviewer #1: The authors present an alternative method for purifying recombinant bacterial collagens (Collagen-like proteins, CLP) containing structural perturbations or more specifically the mutation Gly to Arg. The motivation for developing the purification method is that the widely used purification method, which uses trypsin digestion to remove the affinity tag, also degrade the structural altered CLP. Instead of trypsin, TEV protease and TEV protease cleavage is used for removal of the affinity tag.

In total the authors make four constructs with TEV or Trypsin cleavage sites combined with native CLP or CLP containing the Gly-Arg mutations. By blabla and enzymatic digestion assay, they demonstrate that CLPs containing Gly→Arg mutations are readily digested by trypsin while digestion with TEV protease only cleaved the affinity tag.

The manuscript would benefit from:

• Specify the rationale for choosing to substitute Glycine with Arginine

• Indicate the number of the amino acid substitution or specify which glycine was substituted in the integrin binding site.

• Reference for the contribution of the mutation Gly to Arg change the structure of CLP

• Reference or argument that Gly to X would have same effect as Gly to Arg

• Figure 1:

o Panel C: Missing figure text.

o Panel A:

Exchange “Protease cleavage sites” with “TEV or Trypsin cleavage site”.

Specify which glycine was mutated in the integrin binding site

• Line 79: forgo forgot

Reviewer #2: This manuscript by Gahlawat et al described a simple laboratory effort to purify a collagen like protein (CLP) having Gly-to-X mutations that was first expressed as a fusion protein. They provided evidence showing protease TEV is better suited for the removal of the His-tagged V-domain than trypsin because of the folding problem related to the mutation. Yet, there is no data and no mention on the folding of the peptides. Did the digestion act on monomer during purification or on folded trimer? Can the triple helix form without the removal of V-domain? If not, how can they claim the varied sensitivity of the fusion protein to proteases is related to the ‘partially folded structure’?

Other major problems:

Figure 1 is missing.

The optimization experiments are notoriously difficult to reproduce. The authors did not mention the reproducibility of the results.

Overall, the work felt like an incomplete effort and did not have enough reproducible results to meet the requirement of a research article.

6. PLOS authors have the option to publish the peer review history of their article (what does this mean?). If published, this will include your full peer review and any attached files.

Reviewer #1: No

Reviewer #2: No

<quillbot-extension-portal></quillbot-extension-portal>

---

## [Author Response · Author response to Decision Letter 0]

26 Apr 2023

PONE-D-23-06762

Purification of recombinant bacterial collagens containing structural perturbations

We would like to thank the reviewers for their effort and time in reviewing the manuscript. We sincerely appreciate their valuable and insightful comments, which helped us in improving the quality of the manuscript. We have addressed the comments specific to each reviewer below. We have uploaded this in a formatted version with the added figure to the revised documents, labeled "Response to Reviewers".

Reviewer 1

1. Specify the rationale for choosing to substitute Glycine with Arginine.

Thank you for the suggestion. We have included the rationale for choosing Gly→Arg mutation in lines 83-87.

Lines 83-87:

Structurally distorted triple helices were generated through the introduction of Gly→Arg mutations within the integrin-binding site of CL domains. The Gly→Arg mutation would provide differential folding effects and has been associated with collagen disorders, including Osteogenesis Imperfecta and Ehlers-Danlos Syndrome and its subtypes (https://eds.gene.le.ac.uk/).

2. Indicate the number of the amino acid substitution or specify which glycine was substituted in the integrin binding site.

As listed in lines 120-122, Gly at position #201 and #444 were mutated to Arg residues. Within the integrin-binding site, GFPGER, the second Gly residue (highlighted in bold) was mutated to Arg amino acid, which has further explained in lines 176-178 and Fig 2 legend.

Lines 176-178:

The VCL2(G→R) mutant protein includes a single Gly→Arg mutation (GFPRER) near the middle of the CL domain within the IBS sequence to have minimum impact on the folding and assembly of the triple helix

Fig 2B:

Version of the VCL2 construct with Gly→Arg within the IBS, where the second Gly residue of GFPGER was mutated to Arg (highlighted by “★”).

3. Reference for the contribution of the mutation Gly to Arg change the structure of CLP.

Reference 22 (line 238) supports the impact of Gly→Arg mutation on the triple helix structure. Additional references have been added in support of the argument (28-30). 

4. Reference or argument that Gly to X would have same effect as Gly to Arg.

We have added text in lines 252-256 justifying this claim.

Lines 252-256:

Replacing the trypsin cleavage site with TEV offers a reliable method to isolate CL domains containing structural perturbations irrespective of the type of substituted amino acid, the location of the mutation, and the surrounding local sequence, all of which play a decisive role in determining the conformation and stability of the triple helix.

5. Figure 1:

a. Panel C: Missing figure text.

It is unclear which figure text is missing. The submitted Fig 2C contains a table, which describes the design of protein constructs, including the presence or absence of Gly substitution and the type of protease cleavage site (trypsin vs. TEV).

b. Panel A:

i. Exchange “Protease cleavage sites” with “TEV or Trypsin cleavage site”.

ii. Specify which glycine was mutated in the integrin binding site

i. We have exchanged the “protease cleavage sites” with “TEV or Trypsin cleavage site” in Fig 2A.

ii. We have updated the Fig 2B legend to highlight which Gly residue was mutated to Arg within the integrin-binding site.

Fig 2B:

Version of the VCL2 construct with Gly→Arg within the IBS, where the second Gly residue of GFPGER was mutated to Arg (highlighted by “★”).

6. Line 79: forgo forgot

There is no spelling error in line 79. Because of Gly→X mutations on the triple helix structure, it is challenging to remove an affinity tag and isolate collagen-like domains containing such mutations using trypsin enzyme. For that reason, authors intentionally do not cleave the affinity tag to prevent protein degradation. 

Reviewer 2

1. This manuscript by Gahlawat et al described a simple laboratory effort to purify a collagen like protein (CLP) having Gly-to-X mutations that was first expressed as a fusion protein. They provided evidence showing protease TEV is better suited for the removal of the His-tagged V-domain than trypsin because of the folding problem related to the mutation. 

a. Yet, there is no data and no mention on the folding of the peptides.

We have added circular dichroism (CD) spectra in Supplementary Information (Fig S1) for the different CLPs demonstrating that the constructs properly fold into a triple helix as evidenced by a minimum and a maximum peak around 198-200 nm and 220-225 nm, respectively.

b. Did the digestion act on monomer during purification or on folded trimer?

The trypsin digestion assays were performed on the folded trimers and the digestion profile was visualized using SDS-PAGE. Due to the inclusion of denaturing chemicals and reducing agents, only the monomers will be seen.

c. Can the triple helix form without the removal of V-domain? If not, how can they claim the varied sensitivity of the fusion protein to proteases is related to the ‘partially folded structure’?

Yes, the bacterial collagen triple helix can form without the removal of V-domain. 

2. Figure 1 is missing.

Fig 1 is attached. 

3. The optimization experiments are notoriously difficult to reproduce. The authors did not mention the reproducibility of the results

We agree with the reviewer that optimization experiments are challenging. Using the reagents and chemicals listed in the Materials and Methods section of the manuscript, we obtained similar yields and purity of proteins in multiple batches, thus offering reproducible results. Since multiple factors can impact protein expression and purification, which ultimately affects protein yield and purity, it is always recommended to tailor the expression and purification of each protein construct by screening different culturing conditions. We emphasize that optimization is not the main theme of

---

## [Decision Letter · Decision Letter 1]

3 May 2023

Purification of recombinant bacterial collagens containing structural perturbations

PONE-D-23-06762R1

Dear Dr. Shreiber,

We’re pleased to inform you that your manuscript has been judged scientifically suitable for publication and will be formally accepted for publication once it meets all outstanding technical requirements.

Kind regards,

Yong Wang

Academic Editor

PLOS ONE

Additional Editor Comments (optional):

Reviewers' comments:

Reviewer's Responses to Questions

**Comments to the Author**

1. If the authors have adequately addressed your comments raised in a previous round of review and you feel that this manuscript is now acceptable for publication, you may indicate that here to bypass the “Comments to the Author” section, enter your conflict of interest statement in the “Confidential to Editor” section, and submit your "Accept" recommendation.

Reviewer #1: All comments have been addressed

2. Is the manuscript technically sound, and do the data support the conclusions?

Reviewer #1: Yes

3. Has the statistical analysis been performed appropriately and rigorously? 

Reviewer #1: N/A

4. Have the authors made all data underlying the findings in their manuscript fully available?

Reviewer #1: Yes

5. Is the manuscript presented in an intelligible fashion and written in standard English?

Reviewer #1: Yes

6. Review Comments to the Author

Reviewer #1: All comments have been addressed satisfactorily. I have no additional questions. The use of TEV proteases for purification of CLP has been demonstrated satisfactorily.

7. PLOS authors have the option to publish the peer review history of their article (what does this mean?). If published, this will include your full peer review and any attached files.

Reviewer #1: No

<quillbot-extension-portal></quillbot-extension-portal>

---

## [Editor Report · Acceptance letter]

9 May 2023

PONE-D-23-06762R1 

Purification of recombinant bacterial collagens containing structural perturbations 

Dear Dr. Shreiber:

I'm pleased to inform you that your manuscript has been deemed suitable for publication in PLOS ONE. Congratulations! Your manuscript is now with our production department. 

Kind regards, 

on behalf of

Dr. Yong Wang 

Academic Editor

PLOS ONE